# Determinants of Late HIV Presentation at Ndlavela Health Center in Mozambique

**DOI:** 10.3390/ijerph19084568

**Published:** 2022-04-11

**Authors:** Jeremias Salomão Chone, Ana Barroso Abecasis, Luís Varandas

**Affiliations:** 1Global Health and Tropical Medicine (GHTM), Institute of Hygiene and Tropical Medicine, Universidade Nova de Lisboa (IHMT-UNL), Rua da Junqueira 100, 1349-008 Lisbon, Portugal; ana.abecasis@ihmt.unl.pt (A.B.A.); varandas@ihmt.unl.pt (L.V.); 2Ministry of Health, Av. Eduardo Mondlane 1008, Maputo 264, Mozambique; 3Instituto Superior de Ciências de Saúde, Av. Tomás Nduda, Maputo 3276, Mozambique; 4Faculty of Ethics and Human Science, São Tomás University, Av. Ahmed Sekou Touré 690, Maputo 2088, Mozambique; 5Faculdade de Ciências Médicas, NOVA Medical School, Universidade Nova de Lisboa, Campo Mártires da Pátria 130, 1169-056 Lisbon, Portugal; 6Hospital CUF Descobertas, Rua Mário Botas, 1998-018 Lisbon, Portugal

**Keywords:** late presentation, HIV, Mozambique, determinants

## Abstract

Background: There has been tremendous progress in the fight against HIV worldwide; however, challenges persist in the control of HIV infection. These challenges include the high prevalence of late presenters. There are many disadvantages of late presentation—from reduced survival of the infected person to the risk of transmitting the infection. This research aims to analyze the factors that influence the late presentation in patients attending Ndlavela Health Center in Mozambique. Methodology: A retrospective cross-sectional study was carried out at Ndlavela Health Center including patients diagnosed with HIV between 2015 and 2020. The European Late Presenter Consensus working group definitions were used, and univariate and multivariate logistic regression were used to identify factors associated with late presentation. Results: In total, 519 participants were included in the study, of which nearly 47% were classified as late presenters. The male gender (AOR = 2.41), clinical suspicious test (AOR = 4.03), initiated by the health professional (AOR = 2.1,9), and fear of stigma (AOR = 2.80) were the main risk factors for late HIV presentation. Conclusion: Factors that are potentially determinant for late HIV presentation were identified. Actions are needed to focus on risk factors that are most likely to delay presentation.

## 1. Introduction

It is estimated that in 2019, there were around 38 million people living with HIV worldwide. A large part of this population is living in sub-Saharan Africa, representing 54% of people living with HIV in the world [1,2,3].

According to the Immunization, Malaria, and HIV/AIDS Indicator Survey (IMASIDA), in 2015, Mozambique had an HIV prevalence of 13.2% (15.4% in women and 10.1% in men) [4]. This fact can be associated with the lack of knowledge about the HIV serological status, which contributes to propagated transmission by infected and untreated patients.

Although there is progress in the fight against HIV, some adversities remain in controlling this pandemic. One of the most important is the continuing high prevalence of late presentation of HIV infection, even after the widespread availability of diagnostic tests.

The European Late Presenter Consensus working group defines the late presentation of HIV as a lymphocyte CD4+ count lower than 350 cells/μL at the time of diagnosis or, regardless of the lymphocyte count, presentation with an AIDS-defining event at the time of diagnosis. Very late presentation, on the other hand, is defined as diagnosis with lymphocyte CD4+ count lower than 200 cells/μL or, regardless of the lymphocyte count, presentation with an AIDS-defining event [1,5]. 

There is considerable heterogeneity in definitions of late HIV presentation. The literature review made by Mukolo et al. [6] concluded that there are several criteria (e.g., lymphocyte CD4+ count lower than 200 cells/μL, clinical criteria of AIDS-defining illness, the time interval between HIV diagnosis and progression to AIDS) used by different authors and context to define late HIV presentation. 

However, most of the articles analyzed use immunological criteria to classify late or non-late HIV presentations.

While the estimation of the prevalence of late presentation is hampered by varying definitions, the prevalence of late presentation varies worldwide. However, it is estimated that 13% to 43% of new HIV diagnoses correspond to late presentation worldwide [7]. On the African continent, several studies indicate that the prevalence of late presentation varies, depending on the country, from 40% to 90% [8,9,10,11,12,13,14].

The knowledge of the serological status allows patients to benefit from antiretroviral treatment, reducing the probability of transmitting the infection either by reducing viral load or by changing risk behaviors [15,16].

Despite all pandemic control measures implemented and the widespread availability of antiretroviral treatment, the incidence of HIV remains stable, with increases in some industrialized countries in recent decades, suggesting those who do not know their HIV status contribute to the continued transmission of the virus.

In the United States (US), it is estimated that 25% of HIV patients who are unaware of their HIV status are responsible for 54% of new infections. In Europe, about 30% of HIV patients are unaware of their HIV status [2,17]. 

The impact of late presentation is tremendous, with faster disease progression, increased susceptibility to opportunistic infections, and higher costs to the healthcare provider system. From the public health point of view, it increases the risk of HIV transmission, perpetuating new infections [3,7,10,13,18,19,20,21].

The increasing prevalence of late presentation has a considerable impact on achieving the “95–95–95” goals by 2025. Regarding the first 95, a higher prevalence of late presentation implies the existence of a high number of people who do not know their HIV serological status. Concerning the second 95, the increase in late presentation obviously implies a higher proportion of patients who are not on antiretroviral (ARV) therapy. Finally, for the third 95, late presentation implies a higher probability of therapeutic failure and viral suppression more difficult to achieve [22]. As such, the delay in the use of HIV health services has catastrophic consequences both at the patient level, increasing morbidity and mortality of late presenters, and at the population level, as an impediment to the control of HIV infection. 

This research aims to identify and analyze the factors associated with a late presentation in patients diagnosed at the Ndlavela Health Center, in Maputo, Mozambique, between the years 2015 and 2020. 

## 2. Materials and Methods

### 2.1. Study Design

This was a retrospective cross-sectional study, carried out in Maputo, Mozambique, at Ndlavela Health Center, in patients diagnosed with HIV between 1 January 2015 and 31 December 2020.

The study included 519 patients diagnosed with HIV over 18 years old. The sample was selected using a convenience sampling approach, based on the patients who attended the health facility during the data collection period.

All patients who were below 18 years old, those who did not have a CD4 lymphocyte count record in the patient’s individual record in the first month after the diagnosis, those who did not have a definition of the stage of HIV infection according to the World Health Organization (WHO), those who did not remember the answer in four or more questions, those who were pregnant, and those who did not give consent were excluded. 

The categorization into late and non-late presentations was made according to the classification criteria proposed by the European late Presenter Consensus working group [1,5]. 

### 2.2. Data Collection

A questionnaire was exclusively designed for this study to identify the sociodemographic, behavioral characteristics, and psychosocial factors that may be associated with the late presentation of HIV infection. After the questionnaire was filled out, the medical history of the patients was collected to know the HIV infection stage, comorbidities, and lymphocyte CD4 count at the time of diagnosis. A pilot test was performed with 20 patients who meet the inclusion criteria. This test helped to validate the questionnaire to satisfy the research objective.

### 2.3. Data Analysis

The statistical analyses were performed in the Statistical Package for Social Science (SPSS) version 26. First, we performed the descriptive statistics and median for quantitative variables to describe the sociodemographic and behavioral characteristics of the study participants. Afterward, to assess the association between the different variables (independent variables and the dependent variable) and late presentation, the chi-squared test of independence and Fisher’s exact test were performed. 

The dependent variable was the time of presentation (late or non-late presentation), while the independent variables were the year of diagnosis, gender, age marital status at the time of diagnosis, suggestive symptoms of HIV before the diagnosis, reason for HIV test, prior screened for HIV, fear of stigma, and prior information about HIV.

Predictor variables that were statistically associated with the chi-squared test of independence were selected to compound the logistic regression model. We used bivariate logistic regression models to determine the odds ratio (OR) of late presentation.

To minimize the potential collinearity, we assessed the correlation between all pairs of independent variables and verified that no pair of variables included in the same regression model was highly correlated. To build a multivariate logistic regression model and generate adjusted odds ratios (AORs), to avoid confounding variables, we included the year of diagnosis, gender, age, marital status at the time of diagnosis, symptoms suggestive of HIV infection before the diagnosis, the reason for HIV testing, screening for HIV before HIV diagnosis, fear of stigma, and habit of regular medical checkups before the diagnosis. The results are presented as estimated odds ratio (OR) and adjusted odds ratio (AOR), with corresponding 95% confidence intervals (CIs) and *p*-values. 

The goodness of fit of the model was verified by the Hosmer–Lemeshow, comparing the expected frequencies with the observed frequency (*p* = 0.642), which indicated that the model fits the data well. The model also showed 69% of the overall percentage and Omnibus tests <0.001. All tests were performed with a 0.05 significance level. Temporal trends in the prevalence of late presentation were assessed using a nonparametric test. 

### 2.4. Ethics Approval

This study was submitted for evaluation by the National Bioethics Commission for Health, Ministry of Health of Mozambique, and was approved and registered under number 83/CNBS/2020, respecting the declaration of Helsinki. The anonymity and confidentiality of participants were guaranteed by the investigators at all stages of the study. 

## 3. Results

A total of 519 patients diagnosed with HIV between 1 January 2015 and 31 December 2020 participated in this study. The prevalence of late presentation over the five years was 46.6% (242). Table 1 and Table 2 show the results of sociodemographic, behavioral, and clinical factors of the study participants.

Patients aged between 18 and 84 years old participated in this research, and the median age of participants was 35 years old. More than two-thirds (76.7% (398)) were female, and half of the participants (50.7% (263)) had a primary level of education. More than half (56.8% (295)) were married or living with partners at the time of diagnosis, and the highest proportion indicated to be in monogamous relationships (79.4% (412)) at the time of diagnosis.

The majority (64.9% (337)) did not perform HIV screening tests regularly; however, at the time of diagnosis, the initiative to perform the test was mostly carried out by patients themselves (60.5% (314)), with voluntary counseling test.

Figure 1 shows the prevalence of late presenters over the years. 

The Table 3 show the result of multivariate analysis. Patients diagnosed in 2020 had an 84% lower chance of being late presenters, when compared with patients diagnosed in 2015 (AOR = 0.16; 95% CI 0.07–0.37). Being male was revealed to be a risk factor for late presentation, with a 2.41 higher chance of being a late presenter when compared with women (AOR = 2.41; 95% CI 1.85–4.35).

Adults aged >50 years old were 1.62 times more likely to be diagnosed late than those aged 18–24 years old (AOR = 1.62; 95% CI 0.64–4.06, *p* = 0.303).

Those who were never married or lived with their partners had a 60% lower chance of being late presenters when compared with those who were married or living with their partners (AOR = 0.31; 95% CI 0.13–0.98, *p* = 0.042).

Having had regular checkups before the diagnosis implied a 77% lower risk for a late presentation, compared with not having had regular checkups (AOR = 0.23; 95% CI 0.14–0.65, *p* = 0.04)

The patients who underwent the test by the provider initiative order were 2.19 times more likely to be late presenters when compared with those who performed the test on their own (VCT) (AOR = 2.19; 95% CI 1.87–3.04, *p* = 0.002).

Those who had been diagnosed with symptoms suggestive of infection were four times more likely of late presentation, compared with patients who had no symptoms (AOR = 4; 95% CI 2.6–6.1, *p* > 0.001)

On the other hand, fear of being stigmatized for being HIV positive was identified as a risk factor for late presentation. Patients who had fear of stigma were 2.80 times more likely of being late presenters when compared with those who did not have fear of stigma (AOR = 2.80; 95% CI 1.47–3.36, *p* = 0.042).

## 4. Discussion

To the best of our knowledge, this is the first study in Mozambique to determine the prevalence and explore the determinant factors of the late HIV diagnosis. In recent years, global actions to reduce HIV transmission have intensified. Several programs have been established to promote timely diagnosis of HIV [23].

The knowledge of the prevalence of late presentation can be used as a superior indicator to monitor prevention programs, as well as the effectiveness of testing strategies [12].

In an era in which diagnostic tests for HIV are free and accessible, in this health center, a considerable proportion of late presenters still exists, accounting for 46.6% (242). This prevalence, although lower than the one found in Nigeria, with 85.6% (12,401) [11] and Cameron, with 89.7% (1672) [12], is closer to that found in South Africa, with 59.8% (198) [14].

The prevalence of late presentation over the past years, from 2015 to 2020, had a significant trend of reduction, from 57.5% in 2015 to 20.7% in 2020 *(p* < 0.001) Figure 1.

Patients diagnosed in 2020 and 2019 had, respectively, 85% and 73% lower chance of being late presenters, when compared with those diagnosed in 2015. Since 2016, the Ministry of Health of Mozambique has increased the testing campaigns for HIV. The diagnostic tests started to be offered to all patients who visit the hospital or in the community, which could probably have contributed to the reduced probability of late HIV diagnosis.

The average count of lymphocyte CD4+ cells of patients diagnosed late (244 cells/μL) indicates that they are below the optimum for fast immunological reconstitution; however, these values are higher than those found in Zimbabwe (61 cells/μL) [21] Figure 2.

Most of the study population were female, which could explain the health-seeking behavior disparity between the two genders. Although women are, in Africa, more vulnerable to acquiring HIV, in this study, we found that men were more likely to have late presentations, compared with women. Similar results were found in previous studies carried out in South Africa [19], India [24], sub-Saharan Africa [25], Switzerland [26], Asia [27], Ethiopia [28], Benin [29], and Asia [15].

According to a situational analysis of the African continent, in particular Mozambique, there are no male-oriented HIV control strategies and plans. Plans have been focused on women and children for several years [4,30].

Vertical transmission control strategies, which require mandatory diagnostic tests in antenatal visits, provide more chances to perform testing and thus allow more timely diagnosis in women [14]. Strategies for couple testing and inclusion of men in antenatal and pediatric consultations should probably be strengthened as tools to reduce the likelihood of late presentation in men.

Being single or never having lived with a partner was identified as a protective factor for the HIV late presentation, compared with those who were married or living with partners. There are many possible explanations for this result. Those who are single or never lived with a partner are young adults, so they are aware of the need for frequent diagnostic tests; therefore, when they acquire the infection, they are diagnosed early. On the other hand, those who are married or live with a partner probably believe there is no need for regular testing, but in Mozambique, those who are married or are in a stable relationship contribute 25.6% to HIV incidence [31].

Although the level of education has been identified in previous studies [14,25,26,29,32,33,34] as a factor associated with late HIV diagnosis, in this study, no such association was found (*p* = 0.632).

Having prior information about HIV was a protective factor of late presentation when compared with the counterpart. These results are supported by those found in a study conducted in Ethiopia [9], which concluded that patients with information and a comprehensive understanding of the severity of HIV infection are less likely to be diagnosed late.

Additionally, in Ethiopia, in another study [35], people who presented late had low levels of knowledge about HIV. Patients with medium and high levels of knowledge about HIV were 2 and 3.5 times more likely to be diagnosed early the HIV infection.

Regular diagnostic tests are excellent tools to avoid late presentation. These results reinforce the need for periodic (at least once a year) HIV diagnostic tests, especially in couples and adults who live in “stable” relationships.

Surprisingly, the number of sexual partners at diagnosis, illicit drug use, and regular condom use before HIV-positive results did not have statistically significant associations with late presentation. It is hypothesized that they are not associated with late presentations but only with a greater susceptibility to acquiring HIV. While such patients may be at greater risk to acquire HIV, they may have a higher perception of their risk and therefore test more regularly for HIV.

Stigma is a barrier to the timely diagnosis of HIV. Additionally, lack of social support also contributes strongly to late presentation [9]. As described in previous studies [2,9,19,21,22,35,36,37,38,39], participants who have fear of stigma are 2.80 times more likely to have late presentation when compared with those without a fear of stigma (*p* = 0.042).

In a study in Ethiopia [35], which used the multidimensional measure of internalized HIV stigma [40], patients who had medium-to-high levels of internalized stigma had, respectively, a probability of 4.49 and 16.64 times higher of being late presenters. This suggests that people with high levels of internalized stigma avoid having a diagnostic test and are therefore presenting late.

In Mozambique, communicating to the family HIV positivity and taking ARV can have several negative implications for the patients’ social relationships, such as partner abandonment, family rejection, persecution of relatives, loss of children, and even being banished from their own home.

The patients diagnosed with the infections by the provider-initiated testing were 2.19 times more likely to be late presenters (*p* = 0.002) when compared with those who performed the diagnostic test by voluntary counseling test. Similar results were found in Ethiopia [35], Cabo Verde [8], Cameroon [12], Japan [41], and Zimbabwe [21]. 

Generally, when the test is provider-initiated, there are obvious clinical signs suggestive of the infection, which suggests that the infection evolved in such a way that the patient is already symptomatic.

Contrary to what was found in South Africa [19], the use of traditional medicine did not have a statistically significant association (*p* = 0.310) with late presentation. A plausible explanation for these differences may be due to the knowledge of the participants of this research that traditional medicine is not a panacea. Furthermore, it is well known that traditional medicine does not cure HIV.

This is the first study conducted in Mozambique that depicts the factors associated with late presentation. However, this research has several limitations. First, the variables analyzed are overwhelmingly related to aspects of the patients’ past (before the positive diagnosis). Presumably, there may have been a recall bias. As it is a retrospective cross-sectional study, it is not possible to establish causal relationships. As there are no comparable previous studies conducted in Mozambique, the results must be analyzed with caution and cannot be generalized. However, despite its several limitations, this study provides preliminary information of high value for further research.

## 5. Conclusions

In this study, sociodemographic, behavioral, and psychosocial factors potentially determining the late presentation were identified. It is necessary to reinforce public health intervention actions in the prevention and promotion of timely diagnosis of the infection. These actions will allow reaching population groups with sociodemographic, behavioral, and social characteristics at greater risk for late presentation, as identified in this study. In the absence of prevention campaigns and timely diagnosis, the delay in diagnosis will continue to be an obstacle in the materialization of the control of HIV until 2030.

## Figures and Tables

**Figure 1 ijerph-19-04568-f001:**
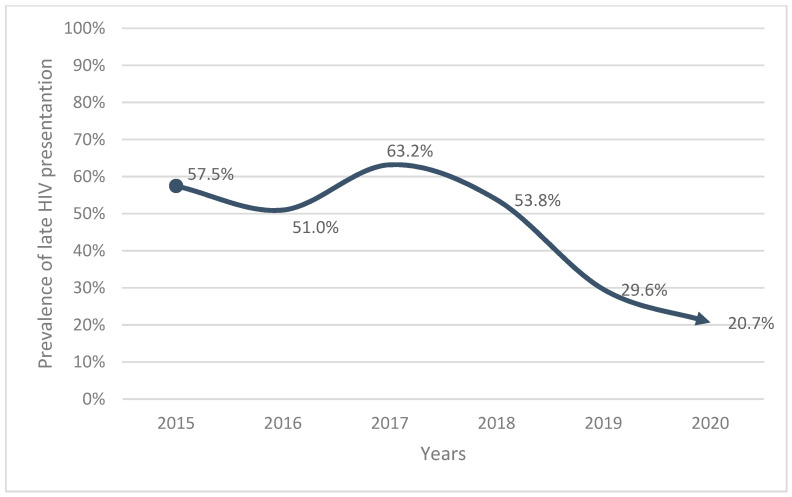
Prevalence of late HIV presentation over the years.

**Figure 2 ijerph-19-04568-f002:**
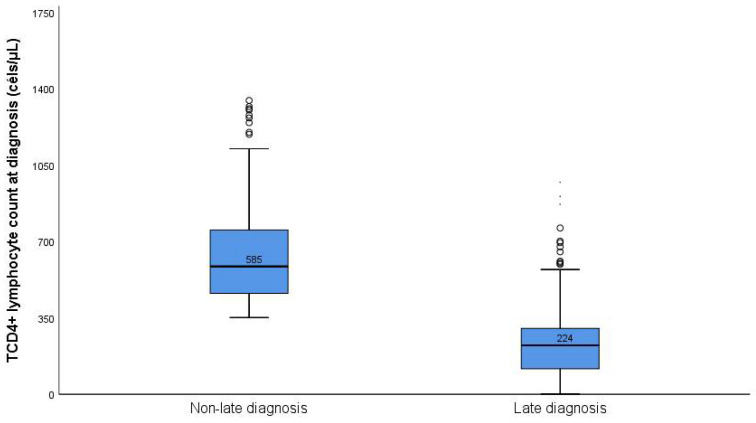
Lymphocyte count at the time of diagnosis.

**Table 1 ijerph-19-04568-t001:** Sociodemographic characteristics of study participants and clinical factors of study participants at the Ndlavela Health Center, Mozambique.

Variables	Total		Late Presentation		Non-Late Presentation		*p* Value *
	*n* 519	(%) 100	*n* 242	(%) 46.6	*n* 277	(%) 53.4	
**Year of diagnosis**							<0.001
2015	80	15.4	46	57.5	34	42.5	
2016	82	15.8	42	51	40	49	
2017	98	18.9	62	63.2	36	36.8	
2018	91	17.5	49	53.8	42	46.2	
2019	91	17.5	27	29.6	64	23.1	
2020	77	14.8	16	20.7	61	22	
**Gender**							0.009
Female	398	76.7	173	71.5	252	81.2	
Male	121	23.3	69	28.5	52	18.8	
**Age**							0.001
18–24	80	15.4	22	9.1	58	20.9	
25–34	176	33.9	79	32.6	97	35	
35–49	211	40.7	113	46.7	98	35.4	
+50	52	10	28	11.6	24	8.7	
**Education level**							0.632
No education	72	13.9	36	14.9	36	13	
Primary	263	50.7	125	51.7	138	49.8	
Secondary or higher	184	35.5	81	33.5	103	37.2	
**Marital status at the time of diagnosis**							0.008
Married or living with a partner	295	56.8	133	55	162	58.5	
Previously married or lived with a partner	75	14.5	47	19.4	28	10.1	
Never married or lived with a partner	149	28.7	62	25.6	87	31.4	

* Chi-squared test for independence.

**Table 2 ijerph-19-04568-t002:** Behavioral factors of study participants at the Ndlavela Health Center, Mozambique.

Variables	Total		Late Presentation		Not Late Presentation		*p* Value *
	** *n* **	**(%)**	** *n* **	**(%)**	** *n* **	**(%)**	
**Some symptoms suggestive of HIV before the diagnosis**							<0.001
No	328	63.2	110	45.5	218	78.7	
Yes	191	36.8	132	54.5	59	21.3	
**Reason for HIV testing**							0.018
VCT	314	60.5	150	62	164	59.2	
PITC	205	39.5	92	38	113	40.8	
**Number of sexual partners at diagnosis**							0.302
None	51	9.8	27	11.2	24	8.7	
Only one	412	79.4	185	76.4	227	81.9	
More than one	56	10.8	30	12.4	26	9.4	
**Prior screened for HIV**							0.009
No	337	64.9	167	69	170	61.4	
Yes	182	35.1	75	31	107	38.6	
**Used illicit drug at the time of diagnosis**							0.482 ψ
No	511	98.5	237	97.9	274	98.9	
Yes	8	1.5	5	2.1	3	1.1	
**Regularly used condoms, before the positive result**							0.668
No	414	79.8	195	80.6	219	79.1	
Yes	105	20.2	47	19.4	58	20.9	
**Fear of stigma**							0.029
No	296	57	149	61.6	147	53.1	
Yes	223	43	93	38.4	130	46.9	
**Prior information about HIV**							0.017
No	30	5.8	14	5.8	16	5.8	
Yes	489	94.2	228	94.2	261	94.2	
**Before the diagnosis performed regular check-ups**							0.023
No	383	73.8	190	78.5	193	69.7	
Yes	136	26.2	52	21.5	84	30.3	
**Visited a traditional medicene**							
No	502	96.7	232	95.9	270	97.5	
Yes	17	3.3	10	4.1	7	2.5	

VCT—voluntary counseling test; PITC—provider-initiated testing and counseling; HIV—human immunodeficiency virus; * chi-squared test for independence; ψ Fisher’s exact.

**Table 3 ijerph-19-04568-t003:** Multivariate results (AOR and OR) for variables associated with late HIV diagnosis at the Ndlavela Health Center, Mozambique.

Explanatory Variables	OR (95% CI)	*p*-Value	AOR (95% CI)	*p*-Value
**Year of diagnosis**				
2015	1		1	
2016	1.776 (1.516–1.942)	0.023	2.717 (1.362–3.421)	0.041
2017	1.273 (1.017–2.329)	0.034	1.182 (1.009–2.295)	0.002
2018	2.862 (1.471–3.580)	0.013	1.857 (0.937–2.682)	0.055
2019	0.312 (0.166–0.586)	0.001	0.279 (0.139–0.563)	0.001
2020	0.194 (0.096–0.393)	0.001	0.169 (0.076–0.375)	0.001
**Gender**				
Feminine	1		1	
Male	1.726 (1.144–2.603)	0.009	2.417 (1.853–4.355)	0.028
**Age**				
18–24	1		1	
25–34	2.147 (1.210–3.811)	0.009	1.738 (0.989–3.362)	0.101
35–49	3.040 (1.736–5.324)	<0.001	1.499 (1.054–2.978)	<0.001
+50	3.076 (1.477–6.405)	0.003	1.620 (0.646–4.064)	0.303
**Marital status at the time of diagnosis**				
Married or living with a partner	1		1	
Previously married or lived with a partner	2.045 (1.214–3.443)	0.007	1.542 (0.835–2.846)	0.067
Never married or lived with a partner	0.868 (0.583–0.996)	0.036	0.318 (0.132–0.893)	0.042
**Some symptoms suggestive of HIV before the diagnosis**				
No	1		1	
Yes	5.434 (3.023–6.504)	<0.001	4.033 (2.604–6.155)	<0.001
**Reason for HIV testing**				
VCT	1			
PITC	3.890 (1.625–4.267)	0.018	2.190 (1.875–3.043)	0.002
**Prior screened for HIV**				
No	1		1	
Yes	0.396 (0.184–0.714)	0.009	0.235 (0.142–0.654)	0.041
**Fear of stigma**				
No	1		1	
Yes	4.417 (2.998–6.011)	0.029	2.808 (1.477–3.369)	0.042
**Prior information about HIV**				
No	1		1	
Yes	0.302 (0.278–0.697)	0.017	0.421 (0.276–0.854)	0.039
**Before the diagnosis, performed regular checkups**				
No	1		1	
Yes	0.490 (0.159–0.984)	0.023	0.585 (0.153–0.826)	0.008

AOR—unadjusted odds ratio; OR—odds ratio; VCT—voluntary counseling test; PITC—provider-initiated testing and counseling; HIV—human immunodeficiency virus.

## Data Availability

The datasets generated during and analyzed during the current study are available from the corresponding author on reasonable request.

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
