# Peer review of "Determinants of Late HIV Presentation at Ndlavela Health Center in Mozambique"

_ijerph, 2022, doi:10.3390/ijerph19084568_

Round 1

Reviewer 1 Report

Dear Authors,

I commend you on a paper that deals with a subject that is of value to the academic community. I find the subject of late presentation of HIV infection both necessary to further intervention as well as needed to further curb the spread of HIV. I have found it, from a social sciences perspective, interesting that stigma of HIV is still so prominent despite the myriad efforts to de-stigmatize HIV and PLHIV. 

May I please offer the following changes for you to address?

Page 1 Line 17: Please change "Methodology: This study is a retrospective.......". Thank you.

Page 2 Line 64. Please change to "............. those who do not know their HIV status........." Thank you.

Page 3 Line 134. Kindly check the source of the reference that has not been found regarding the registry of the study with the National Bioethics Commission. 

Page 8 Line 163. Please change to "had a primary level of education". Thank you.

PAGE 8 Lines 164-165: Please change to "indicated to be in monogamous relationships". Thank you.

Page 8 Line 182: It would be much clearer if you stated "Patients who have a fear of stigma were 2.80 times more likely to be a late presenter......." Thank you. 

Page 9 Line 187: Please change to "the determinant factors of late HIV diagnosis". This is much clearer in terms of message conveyance. 

Page 9 Lines 201 to 2014 are unclear. I am not sure what is the meaning of these sentences. Perhaps the authors would like to re-write or re-organized the information in these lines so that their message is clearer? Thank you. 

Page 9 Line 210 is superfluous and adds no meaning to the argument. Please re-write to merge this sentence with the former sentence to create a plausible meaning. Thank you. 

Please be sure to consistently use a "." to denote a point between two numerals. There is a use of a comma in line 230 and this creates some misreading of the text. Please correct this. Thank you.

Line 239: Please change to "people who presented late". Thank you.

Line 262: What do the authors mean by "the house"? Do the authors mean "loss of the house/home"? Please change to be more exact. Thank you.

Line 278: "of the patients' past". Please change.

Overall, your submission is an interesting one and I enjoyed learning from it. In future, it would be good to read a paper on how the stigmatization of testing and HIV are created by socio-cultural factors. Perhaps you may want to consider this idea for future work?

Quick thought: As the journal focuses also on environmental research, how would you address the influence of the environment on this particular study? Thank you. 

Thank you. 

Author Response

Dear Authors,

I commend you on a paper that deals with a subject that is of value to the academic community. I find the subject of late presentation of HIV infection both necessary to further intervention as well as needed to further curb the spread of HIV. I have found it, from a social sciences perspective, interesting that stigma of HIV is still so prominent despite the myriad efforts to de-stigmatize HIV and PLHIV. 

May I please offer the following changes for you to address?

Page 1 Line 17: Please change "Methodology: This study is a retrospective.......". Thank you.

Change done. Thank you

Page 2 Line 64. Please change to "............. those who do not know their HIV status........." Thank you.

Page 3 Line 134. Kindly check the source of the reference that has not been found regarding the registry of the study with the National Bioethics Commission. 

It was an error. We did the cross reference while the registry with the National Bioethics Commission has already been removed out and has been referenced as an attachment. Thank you

Page 8 Line 163. Please change to "had a primary level of education". Thank you.

Change done. Thank you

PAGE 8 Lines 164-165: Please change to "indicated to be in monogamous relationships". Thank you.

Change done. Thank you

Page 8 Line 182: It would be much clearer if you stated "Patients who have a fear of stigma were 2.80 times more likely to be a late presenter......." Thank you. 

Change done. Thank you

Page 9 Line 187: Please change to "the determinant factors of late HIV diagnosis". This is much clearer in terms of message conveyance. 

Change done. Thank you

Page 9 Lines 201 to 2014 are unclear. I am not sure what is the meaning of these sentences. Perhaps the authors would like to re-write or re-organized the information in these lines so that their message is clearer? Thank you. 

We re-write the information in these lines (201 to 214). We hope that the message is now clear.  Thank you

Page 9 Line 210 is superfluous and adds no meaning to the argument. Please re-write to merge this sentence with the former sentence to create a plausible meaning. Thank you. 

Sentence re-writhed. We hope that the message has a plausible meaning now.  Thank you

Please be sure to consistently use a "." to denote a point between two numerals. There is a use of a comma in line 230 and this creates some misreading of the text. Please correct this. Thank you.

Change done. Thank you

Line 239: Please change to "people who presented late". Thank you.

Change done. Thank you

Line 262: What do the authors mean by "the house"? Do the authors mean "loss of the house/home"? Please change to be more exact. Thank you.

By the house the authors mean loss of the home. The authors re-writhed the information to be more clear about the meaning of “house”

Line 278: "of the patients' past". Please change.

Change done. Thank you

Overall, your submission is an interesting one and I enjoyed learning from it. In future, it would be good to read a paper on how the stigmatization of testing and HIV are created by socio-cultural factors. Perhaps you may want to consider this idea for future work?

In fact, there are research project aim to analyze why stigma is still a barrier to early diagnosis in PLHIV using mixed methods approach to obtain more reliable results. This research project has already been submitted to the National Ethics Committee. Thank you

Quick thought: As the journal focuses also on environmental research, how would you address the influence of the environment on this particular study? Thank you. 

The authors believe that the environment in which each patient is inserted influences their behavior regarding HIV, especially about the time of diagnosis.

Thank you. 

The authors are most grateful for the interest reading this research. We also thank you for the recommendations and suggestions. They were all considered to improve this research. Thank you.

Please see the attachment with the suggestion made.

Jeremias Chone, Ana Abecassis and Luís Varandas

Reviewer 2 Report

Determinants of Late HIV Presentation at Ndlavela Health  Center in Mozambique

The authors present a good research work with a topic of crucial importance to achieve “95-95-95” goals of UNAIDS for the control and early diagnosis of HIV in the fight against it.

The objective of the manuscript is to identify and analyze the factors associated with late diagnosis of HIV infection in patients diagnosed at the Ndlavela Health Center, in Maputo, Mozambique, between the years 2015 and 2020.

Despite the above, some issues, which I detail more specifically the following could improve the quality of this manuscript for academic publication:

Abstract

Line 16

Better “late diagnosis of HIV infection”, sometimes “late presentation” is used and other times “late diagnosis”, it is necessary to unify terms throughout the entire text and perhaps the most correct form is “late diagnosis”, although It is important to homogenize the terms

Introduction

Line 73-79

Given that these were the targets for the year 2020, and we are currently in the year 2022, it would be more convenient to replace this explanation with the most up-to-date version of “95-95-95” for the year 2025 from UNAIDS and reference it.

Materials and Methods

Line 97

It is not well understood what they are reformulating with this, please reformulate the sentence

Data collection

Line 102

Put here which is the dependent variable and which are the independent ones, that is, specify each of them.

On the other hand, the questionnaire referred to should be detailed a little more, that is, was it a questionnaire designed exclusively for this study? Was it previously designed? Was it validated or not? If so, what was Cronbach's alpha?

Ethical approval

Line 134

 “Error: Reference source not found”, correct this please

Results

Line 137

Start this section with the explanation of each of the tables and then put an image of them

Line 138-139

“Associated factors”, what factors are these? The most correct would be to put “sociodemographics characteristics and clinical factors of study participants…”

Line 139

Table 1

Do not put "n" in the title of the table

“p value”, prevent the word from being cut off, extend the table more

Line 141-142

Table 2

Idem to the above, do not put "n" in the title of the table and put this table at the same height as the previous one

"Before diagnosis, did you have any ...", do not put the statement of the variables as if they were the questions of the questionnaire, do not put it in an interrogative but: "Some symptoms suggestive of HIV before the diagnosis".

“Before being diagnosis….”, idem to the above

“Before diagnosis, did you…”, idem to the above

Line 145-146

Figure 1

It is convenient when tables and/or figures appear to make a brief description of them that accompanies the image

Line 147-148

Figure 2

Idem to the above

Line 149

Table 3

“AOR and OR”, put these terms at the foot of the table and not in the title of the same

Variable “Before diagnosis, did you have any symptoms…? Idem to what was commented for table 2

Variable “Before being diagnosis…, idem to the above

Variable ““Before diagnosis, did you have any prior…”, idem to the above

Line 153-146

It is better not to put these data at the foot of the table and reflect them in data analysisLine

Line 159-160

The noun is missing here, it is necessary to put here “sociodemographics, bahavioral and social factors”, and the same for Tables 1 and 2.

Line 170-171

AOR= 9.16; 95% CI (0.007-0.37): Where do these data appear in Table 3?

Line 173

In Table 3, AOR=1.49 is for the age range from 35 to 49 years, not for those over 50, please clarify this

Line 184

In addition to all of the above, reference should be made to the rest of the factors associated with late diagnosis that appear in Table 3, such as, for example, having had regular check-ups before the diagnosis implies a lower risk for a late diagnosis compared to not having had it done.

Discussion

Line 194-195

“…of new diagnosis. This prevalence…”

If we talk about new diagnoses, we would be talking about incidence and not prevalence, which is what it says in the text.

Line 271-272

“the use of traditional medicine did not have statistically association (p=0.310) with late diagnosis of HIV”

Where is this data observed in the tables of the manuscript?

Line 276-282

Another possible limitation of this study would be that, as it is a retrospective cross-sectional study, it is not possible to establish causal relationships.

Determinants of Late HIV Presentation at Ndlavela Health  Center in Mozambique

The authors present a good research work with a topic of crucial importance to achieve “95-95-95” goals of UNAIDS for the control and early diagnosis of HIV in the fight against it.

The objective of the manuscript is to identify and analyze the factors associated with late diagnosis of HIV infection in patients diagnosed at the Ndlavela Health Center, in Maputo, Mozambique, between the years 2015 and 2020.

Despite the above, some issues, which I detail more specifically the following could improve the quality of this manuscript for academic publication:

Abstract

Line 16

Better “late diagnosis of HIV infection”, sometimes “late presentation” is used and other times “late diagnosis”, it is necessary to unify terms throughout the entire text and perhaps the most correct form is “late diagnosis”, although It is important to homogenize the terms

Introduction

Line 73-79

Given that these were the targets for the year 2020, and we are currently in the year 2022, it would be more convenient to replace this explanation with the most up-to-date version of “95-95-95” for the year 2025 from UNAIDS and reference it.

Materials and Methods

Line 97

It is not well understood what they are reformulating with this, please reformulate the sentence

Data collection

Line 102

Put here which is the dependent variable and which are the independent ones, that is, specify each of them.

On the other hand, the questionnaire referred to should be detailed a little more, that is, was it a questionnaire designed exclusively for this study? Was it previously designed? Was it validated or not? If so, what was Cronbach's alpha?

Ethical approval

Line 134

 “Error: Reference source not found”, correct this please

Results

Line 137

Start this section with the explanation of each of the tables and then put an image of them

Line 138-139

“Associated factors”, what factors are these? The most correct would be to put “sociodemographics characteristics and clinical factors of study participants…”

Line 139

Table 1

Do not put "n" in the title of the table

“p value”, prevent the word from being cut off, extend the table more

Line 141-142

Table 2

Idem to the above, do not put "n" in the title of the table and put this table at the same height as the previous one

"Before diagnosis, did you have any ...", do not put the statement of the variables as if they were the questions of the questionnaire, do not put it in an interrogative but: "Some symptoms suggestive of HIV before the diagnosis".

“Before being diagnosis….”, idem to the above

“Before diagnosis, did you…”, idem to the above

Line 145-146

Figure 1

It is convenient when tables and/or figures appear to make a brief description of them that accompanies the image

Line 147-148

Figure 2

Idem to the above

Line 149

Table 3

“AOR and OR”, put these terms at the foot of the table and not in the title of the same

Variable “Before diagnosis, did you have any symptoms…? Idem to what was commented for table 2

Variable “Before being diagnosis…, idem to the above

Variable ““Before diagnosis, did you have any prior…”, idem to the above

Line 153-146

It is better not to put these data at the foot of the table and reflect them in data analysisLine

Line 159-160

The noun is missing here, it is necessary to put here “sociodemographics, bahavioral and social factors”, and the same for Tables 1 and 2.

Line 170-171

AOR= 9.16; 95% CI (0.007-0.37): Where do these data appear in Table 3?

Line 173

In Table 3, AOR=1.49 is for the age range from 35 to 49 years, not for those over 50, please clarify this

Line 184

In addition to all of the above, reference should be made to the rest of the factors associated with late diagnosis that appear in Table 3, such as, for example, having had regular check-ups before the diagnosis implies a lower risk for a late diagnosis compared to not having had it done.

Discussion

Line 194-195

“…of new diagnosis. This prevalence…”

If we talk about new diagnoses, we would be talking about incidence and not prevalence, which is what it says in the text.

Line 271-272

“the use of traditional medicine did not have statistically association (p=0.310) with late diagnosis of HIV”

Where is this data observed in the tables of the manuscript?

Line 276-282

Another possible limitation of this study would be that, as it is a retrospective cross-sectional study, it is not possible to establish causal relationships.

Determinants of Late HIV Presentation at Ndlavela Health  Center in Mozambique

The authors present a good research work with a topic of crucial importance to achieve “95-95-95” goals of UNAIDS for the control and early diagnosis of HIV in the fight against it.

The objective of the manuscript is to identify and analyze the factors associated with late diagnosis of HIV infection in patients diagnosed at the Ndlavela Health Center, in Maputo, Mozambique, between the years 2015 and 2020.

Despite the above, some issues, which I detail more specifically the following could improve the quality of this manuscript for academic publication:

Abstract

Line 16

Better “late diagnosis of HIV infection”, sometimes “late presentation” is used and other times “late diagnosis”, it is necessary to unify terms throughout the entire text and perhaps the most correct form is “late diagnosis”, although It is important to homogenize the terms

Introduction

Line 73-79

Given that these were the targets for the year 2020, and we are currently in the year 2022, it would be more convenient to replace this explanation with the most up-to-date version of “95-95-95” for the year 2025 from UNAIDS and reference it.

Materials and Methods

Line 97

It is not well understood what they are reformulating with this, please reformulate the sentence

Data collection

Line 102

Put here which is the dependent variable and which are the independent ones, that is, specify each of them.

On the other hand, the questionnaire referred to should be detailed a little more, that is, was it a questionnaire designed exclusively for this study? Was it previously designed? Was it validated or not? If so, what was Cronbach's alpha?

Ethical approval

Line 134

 “Error: Reference source not found”, correct this please

Results

Line 137

Start this section with the explanation of each of the tables and then put an image of them

Line 138-139

“Associated factors”, what factors are these? The most correct would be to put “sociodemographics characteristics and clinical factors of study participants…”

Line 139

Table 1

Do not put "n" in the title of the table

“p value”, prevent the word from being cut off, extend the table more

Line 141-142

Table 2

Idem to the above, do not put "n" in the title of the table and put this table at the same height as the previous one

"Before diagnosis, did you have any ...", do not put the statement of the variables as if they were the questions of the questionnaire, do not put it in an interrogative but: "Some symptoms suggestive of HIV before the diagnosis".

“Before being diagnosis….”, idem to the above

“Before diagnosis, did you…”, idem to the above

Line 145-146

Figure 1

It is convenient when tables and/or figures appear to make a brief description of them that accompanies the image

Line 147-148

Figure 2

Idem to the above

Line 149

Table 3

“AOR and OR”, put these terms at the foot of the table and not in the title of the same

Variable “Before diagnosis, did you have any symptoms…? Idem to what was commented for table 2

Variable “Before being diagnosis…, idem to the above

Variable ““Before diagnosis, did you have any prior…”, idem to the above

Line 153-146

It is better not to put these data at the foot of the table and reflect them in data analysisLine

Line 159-160

The noun is missing here, it is necessary to put here “sociodemographics, bahavioral and social factors”, and the same for Tables 1 and 2.

Line 170-171

AOR= 9.16; 95% CI (0.007-0.37): Where do these data appear in Table 3?

Line 173

In Table 3, AOR=1.49 is for the age range from 35 to 49 years, not for those over 50, please clarify this

Line 184

In addition to all of the above, reference should be made to the rest of the factors associated with late diagnosis that appear in Table 3, such as, for example, having had regular check-ups before the diagnosis implies a lower risk for a late diagnosis compared to not having had it done.

Discussion

Line 194-195

“…of new diagnosis. This prevalence…”

If we talk about new diagnoses, we would be talking about incidence and not prevalence, which is what it says in the text.

Line 271-272

“the use of traditional medicine did not have statistically association (p=0.310) with late diagnosis of HIV”

Where is this data observed in the tables of the manuscript?

Line 276-282

Another possible limitation of this study would be that, as it is a retrospective cross-sectional study, it is not possible to establish causal relationships.

Determinants of Late HIV Presentation at Ndlavela Health  Center in Mozambique

The authors present a good research work with a topic of crucial importance to achieve “95-95-95” goals of UNAIDS for the control and early diagnosis of HIV in the fight against it.

The objective of the manuscript is to identify and analyze the factors associated with late diagnosis of HIV infection in patients diagnosed at the Ndlavela Health Center, in Maputo, Mozambique, between the years 2015 and 2020.

Despite the above, some issues, which I detail more specifically the following could improve the quality of this manuscript for academic publication:

Abstract

Line 16

Better “late diagnosis of HIV infection”, sometimes “late presentation” is used and other times “late diagnosis”, it is necessary to unify terms throughout the entire text and perhaps the most correct form is “late diagnosis”, although It is important to homogenize the terms

Introduction

Line 73-79

Given that these were the targets for the year 2020, and we are currently in the year 2022, it would be more convenient to replace this explanation with the most up-to-date version of “95-95-95” for the year 2025 from UNAIDS and reference it.

Materials and Methods

Line 97

It is not well understood what they are reformulating with this, please reformulate the sentence

Data collection

Line 102

Put here which is the dependent variable and which are the independent ones, that is, specify each of them.

On the other hand, the questionnaire referred to should be detailed a little more, that is, was it a questionnaire designed exclusively for this study? Was it previously designed? Was it validated or not? If so, what was Cronbach's alpha?

Ethical approval

Line 134

 “Error: Reference source not found”, correct this please

Results

Line 137

Start this section with the explanation of each of the tables and then put an image of them

Line 138-139

“Associated factors”, what factors are these? The most correct would be to put “sociodemographics characteristics and clinical factors of study participants…”

Line 139

Table 1

Do not put "n" in the title of the table

“p value”, prevent the word from being cut off, extend the table more

Line 141-142

Table 2

Idem to the above, do not put "n" in the title of the table and put this table at the same height as the previous one

"Before diagnosis, did you have any ...", do not put the statement of the variables as if they were the questions of the questionnaire, do not put it in an interrogative but: "Some symptoms suggestive of HIV before the diagnosis".

“Before being diagnosis….”, idem to the above

“Before diagnosis, did you…”, idem to the above

Line 145-146

Figure 1

It is convenient when tables and/or figures appear to make a brief description of them that accompanies the image

Line 147-148

Figure 2

Idem to the above

Line 149

Table 3

“AOR and OR”, put these terms at the foot of the table and not in the title of the same

Variable “Before diagnosis, did you have any symptoms…? Idem to what was commented for table 2

Variable “Before being diagnosis…, idem to the above

Variable ““Before diagnosis, did you have any prior…”, idem to the above

Line 153-146

It is better not to put these data at the foot of the table and reflect them in data analysisLine

Line 159-160

The noun is missing here, it is necessary to put here “sociodemographics, bahavioral and social factors”, and the same for Tables 1 and 2.

Line 170-171

AOR= 9.16; 95% CI (0.007-0.37): Where do these data appear in Table 3?

Line 173

In Table 3, AOR=1.49 is for the age range from 35 to 49 years, not for those over 50, please clarify this

Line 184

In addition to all of the above, reference should be made to the rest of the factors associated with late diagnosis that appear in Table 3, such as, for example, having had regular check-ups before the diagnosis implies a lower risk for a late diagnosis compared to not having had it done.

Discussion

Line 194-195

“…of new diagnosis. This prevalence…”

If we talk about new diagnoses, we would be talking about incidence and not prevalence, which is what it says in the text.

Line 271-272

“the use of traditional medicine did not have statistically association (p=0.310) with late diagnosis of HIV”

Where is this data observed in the tables of the manuscript?

Line 276-282

Another possible limitation of this study would be that, as it is a retrospective cross-sectional study, it is not possible to establish causal relationships.

Author Response

Determinants of Late HIV Presentation at Ndlavela Health  Center in Mozambique

The authors present a good research work with a topic of crucial importance to achieve “95-95-95” goals of UNAIDS for the control and early diagnosis of HIV in the fight against it.

The objective of the manuscript is to identify and analyze the factors associated with late diagnosis of HIV infection in patients diagnosed at the Ndlavela Health Center, in Maputo, Mozambique, between the years 2015 and 2020.

Despite the above, some issues, which I detail more specifically the following could improve the quality of this manuscript for academic publication:

Abstract

Line 16

Better “late diagnosis”, sometimes “late presentation” is used and other times “late diagnosis”, it is necessary to unify terms throughout the entire text and perhaps the most correct form is “late diagnosis”, although It is important to homogenize the terms

Terms homogenized. The authors prefer to use the term “late presentation”. Thank you

Introduction

Line 73-79

Given that these were the targets for the year 2020, and we are currently in the year 2022, it would be more convenient to replace this explanation with the most up-to-date version of “95-95-95” for the year 2025 from UNAIDS and reference it.

Change done. Thank you

Materials and Methods

Line 97

It is not well understood what they are reformulating with this, please reformulate the sentence

Exclusion criteria re-writhed. The authors hope that it is now clear. Thank you

Data collection

Line 102

Put here which is the dependent variable and which are the independent ones, that is, specify each of them.

Recommendation done, but the authors believe this information fits better on the section “data analysis”. Line 118 to 121.

On the other hand, the questionnaire referred to should be detailed a little more, that is, was it a questionnaire designed exclusively for this study? Was it previously designed? Was it validated or not? If so, what was Cronbach's alpha?

Yes, the questionnaire was designed exclusively for this study. Validation and adjustment of the questionnaire trough a pilot test

Ethical approval

Line 134

 “Error: Reference source not found”, correct this please

Correction done. Thank you

Results

Line 137

Start this section with the explanation of each of the tables and then put an image of them

Recommendation done. Thank you

Line 138-139

“Associated factors”, what factors are these? The most correct would be to put “sociodemographics characteristics and clinical factors of study participants…”

Change done. Thank you

Line 139

Table 1

Do not put "n" in the title of the table

“p value”, prevent the word from being cut off, extend the table more

Changes done. Thank you

Line 141-142

Table 2

Idem to the above, do not put "n" in the title of the table and put this table at the same height as the previous one

"Before diagnosis, did you have any ...", do not put the statement of the variables as if they were the questions of the questionnaire, do not put it in an interrogative but: "Some symptoms suggestive of HIV before the diagnosis".

“Before being diagnosis….”, idem to the above

“Before diagnosis, did you…”, idem to the above

Changes done. Thank you

Line 145-146

Figure 1

It is convenient when tables and/or figures appear to make a brief description of them that accompanies the image

Recommendation done. Thank you

Line 147-148

Figure 2

Idem to the above

Line 149

Table 3

“AOR and OR”, put these terms at the foot of the table and not in the title of the same

Variable “Before diagnosis, did you have any symptoms…? Idem to what was commented for table 2

Variable “Before being diagnosis…, idem to the above

Variable ““Before diagnosis, did you have any prior…”, idem to the above

 Recommendation done. Thank you

Line 153-146

It is better not to put these data at the foot of the table and reflect them in data analysis Line

Recommendation done. Thank you

Line 159-160

The noun is missing here, it is necessary to put here “sociodemographics, bahavioral and social factors”, and the same for Tables 1 and 2.

Recommendation done. Thank you

Line 170-171

AOR= 9.16; 95% CI (0.007-0.37): Where do these data appear in Table 3?

There was a mistake. This data do not appear in table 3. The authors wrongly writhed the data above.

Line 173

In Table 3, AOR=1.49 is for the age range from 35 to 49 years, not for those over 50, please clarify this

Changes done. Thank you

Line 184

In addition to all of the above, reference should be made to the rest of the factors associated with late diagnosis that appear in Table 3, such as, for example, having had regular check-ups before the diagnosis implies a lower risk for a late diagnosis compared to not having had it done.

Recommendation done. Initially, the authors prefer not to describe all the variables because they think it would exaggerate the amount of information. Thank you

Discussion

Line 194-195

“…of new diagnosis. This prevalence…”

If we talk about new diagnoses, we would be talking about incidence and not prevalence, which is what it says in the text.

Change done. Thank you

Line 271-272

“the use of traditional medicine did not have statistically association (p=0.310) with late diagnosis of HIV”

Where is this data observed in the tables of the manuscript?

This data unfortunately is not observed in the tables, because the authors did not use in the paper the latest table which have the data above. The authors already uploaded the table with the missing data about traditional medicine. Thank you

Line 276-282

Another possible limitation of this study would be that, as it is a retrospective cross-sectional study, it is not possible to establish causal relationships.

Recommendation done. Thank you

Please see the attachment with the suggestion made.

The authors are most grateful for the interest reading this research. We also thank you for the recommendations and suggestions. They were all considered to improve this research. Thank you.

Jeremias Chone, Ana Abecassis and Luís Varandas
